# Morehouse Choice Accountable Care Organization and Education System (MCACO-ES): Integrated Model Delivering Equitable Quality Care

**DOI:** 10.3390/ijerph16173084

**Published:** 2019-08-25

**Authors:** Michelle Brown, Elizabeth O. Ofili, Debbie Okirie, Priscilla Pemu, Cheryl Franklin, Yoon Suk, Alexander Quarshie, Mohamed Mubasher, Charles Sow, Valerie Montgomery Rice, David Williams, Michael Brooks, Ernest Alema-Mensah, Dominic Mack, Daniel Dawes

**Affiliations:** 1Project Management Office, Morehouse Choice ACO-ES, 1046 Ridgeview Avenue, Suite 4, Atlanta, GA 30315, USA; 2Department of Medicine, Morehouse School of Medicine, 720 Westview drive SW, Atlanta, GA 30310, USA; 3Medical Director’s Office, Morehouse Healthcare, 1800 Howell Mill Road, NW, Atlanta, GA 30318, USA; 4Department of Community Health and Preventive Medicine, Morehouse School of Medicine, 720 Westview Drive SW, Atlanta, GA 30310, USA; 5Department of Family Medicine, Morehouse School of Medicine, 720 Westview Drive, SW, Atlanta, GA 30310, USA; 6Administration, Morehouse School of Medicine and Morehouse Healthcare, 720 Westview Drive, SW, Atlanta, GA 30310, USA; 7Administration, Southside Medical Center, 1046 Ridgeview Avenue, Suite 4, Atlanta, GA 30315, USA; 8Administration, The Family Health Centers of Georgia, 868 York Avenue, Atlanta, GA 30310, USA; 9National Center for Primary Care, Morehouse School of Medicine, 720 Westview Drive SW, Atlanta, GA 30310, USA; 10Satcher Health Leadership Institute, Morehouse School of Medicine, 720 Westview Drive SW, Atlanta, GA 30310, USA

**Keywords:** accountable care organization (ACO), care coordination, health care innovation, value-based performance, Medicare Shared Savings Program (MSSP), Safety-net ACOs

## Abstract

Accountable Care Organizations (ACOs) seek sustainable innovation through the testing of new care delivery methods that promote shared goals among value-based health care collaborators. The Morehouse Choice Accountable Care Organization and Education System (MCACO-ES), or (M-ACO) is a physician led integrated delivery model participating in the Medicare Shared Savings Program (MSSP) offered through the Centers for Medicare and Medicaid Services (CMS) Innovation Center. The MSSP establishes incentivized, performance-based payment models for qualifying health care organizations serving traditional Medicare beneficiaries that promote collaborative efficiency models designed to mitigate fragmented and insufficient access to health care, reduce unnecessary cost, and improve clinical outcomes. The M-ACO integration model is administered through participant organizations that include a multi-site community based academic practice, independent physician practices, and federally qualified health center systems (FQHCs). This manuscript aims to present a descriptive and exploratory assessment of health care programs and related innovation methods that validate M-ACO as a reliable simulator to implement, evaluate, and refine M-ACO’s integration model to render value-based performance outcomes over time. A part of the research approach also includes early outcomes and lessons learned advancing the framework for ongoing testing of M-ACO’s integration model across independently owned, rural, and urban health care locations that predominantly serve low-income, traditional Medicare beneficiaries, (including those who also qualify for Medicaid benefits (also referred to as “dual eligibles”). M-ACO seeks to determine how integration potentially impacts targeted performance results. As a simulator to test value-based innovation and related clinical and business practices, M-ACO uses enterprise-level data and advanced analytics to measure certain areas, including: 1) health program insight and effectiveness; 2) optimal implementation process and workflows that align primary care with specialists to expand access to care; 3) chronic care management/coordination deployment as an effective extender service to physicians and patients risk stratified based on defined clinical and social determinant criteria; 4) adoption of technology tools for patient outreach and engagement, including a mobile application for remote biometric monitoring and telemedicine; and 5) use of structured communication platforms that enable practitioner engagement and ongoing training regarding the shift from volume to value-based care delivery.

## 1. Introduction

National health expenditures are projected to grow at an average annual rate of 5.5 percent for 2018–2027 and represent 19.4 percent of gross domestic product in 2027. It is estimated that more than 95% of the trillions of dollars spent on health care in the United States each year funds direct medical services, even though 60% of preventable deaths are rooted in modifiable behaviors and exposures that occur in the community [1]. Effective coordination of health care, social services, public health, and community-based organizations could improve population health outcomes and advance health equity [2,3,4]. Some encouraging innovations are emerging, catalyzed in part by payers, delivery system reform, and the growth of value-based or shared-risk payment models, to support high-value community focused interventions. However, developing sustainable payment models to support such partnerships remain a challenge [5,6,7].

The Center for Medicare and Medicaid Services (CMS) is testing Accountable Care Organizations (ACOs) and shared savings models as part of its health care innovation program [8,9,10]. The Morehouse Choice Accountable Care Organization and Education System (MCACO-ES), or (M-ACO) is a physician-led, CMS Medicare Shared Savings Program (MSSP) ACO, that has developed and deployed an evolving model for clinical and operational integration among otherwise independently owned health care organizations, rendering primary care, specialty care other community-focused services to urban and rural populations in Georgia. Although committed to the development of an accountable care organization, the collaborators maintain a unique position to extend the definition in a manner that:Builds on the history, mission and value proposition for unparalleled community health improvement of M-ACO and its partners.Incorporates its distinguished training capacity to expand knowledge and understanding about value-based care to clinical care and health administration teams.Optimizes existing and new collaborative community relationships.Makes use of comprehensive, aggregated data and vast experience with underserved populations.Addresses health disparities and social determinants of health.

M-ACO’s focus on measurable population and community health outcomes has a health equity lens that targets intervention of upstream factors impacting healthcare delivery (socioeconomic and social determinants), as well as individual factors (behavior; physiologic markers of disease, e.g. blood pressure; blood glucose). This model adopts the “Triple Aim” which strives to simultaneously improve population health, improve the patient experience of care, and reduce per capita cost. The Institute for Healthcare Improvement (IHI) developed the Triple Aim framework, and it has since become the organizing framework for the National Quality Strategy of the US Department of Health and Human Services (HHS) and for strategies of other public and private health organizations such as the CMS [11] 

The primary objective of this paper is to present our research approach, early outcomes, lessons learned, and future directions for this physician-led and community-focused ACO. Supported by a comparatively larger body of literature, many ACOs represent large, hospital-led health systems, including those that “integrate” small group practices, often through purchase acquisition and other consolidation, whose health care practitioners then become employees of the large health system. Little or no research has been done on care delivery models that bring together independent safety-net, community-focused health care provider groups, enabled by robust centralized, communication and health technology platforms. Our early outcomes demonstrate key performance indicators related to care accessibility, clinical and quality outcomes improvement and earned shared savings delivered through programs that prioritize value-based interventions to high risk patients with multiple chronic conditions. The M-ACO participant organizations align in their innovation goals that continue to drive relevant and timely research in population health and advanced alternative payment reform models. 

## 2. Materials and Methods

### 2.1. M-ACO Participant Organizations

M-ACO participant organizations align on a collective mission-critical approach to data aggregation analysis and actionable intent to deliver high quality, equitable and cost efficient care, through community and school-based practice locations within preexisting evidence-based models as well as new care delivery models designed by the M-ACO. Table 1 shows the list of M-ACO participant organizations that serve 169 rural and urban ambulatory practice locations primarily comprised of health care safety net organizations, and three mobile medical units with over 700 providers (of which 500 are primary care clinicians, including physicians, nurse practitioners and physician assistants)

### 2.2. Program Goals and Objectives

The program goals and objectives of M-ACO resonate with state and local health policy and healthcare reform intentions:Re-design Patient Care to attain “Triple Aim” goals [11]Research and engagement of providers and patients in predominantly underserved communitiesEducation and Training utilizing scalable, digital training modelsCommunity Health bridging biopsychosocial determinants to health outcomesPayment Reform with Aligned Incentives across independent health care organizations

### 2.3. Demographics of M-ACO Service Area

In January 2018, the M-ACO engaged new partnerships and achieved over a 5-fold increase in its service area footprint from 33 ambulatory care health care centers to 169 rural and urban ambulatory locations, expanding its service role in the health care safety-net, moving M-ACO’s total MSSP Medicare beneficiary attribution as appointed by CMS from approximately 5000 to 10,000. Current attribution as of this writing is approximately 15000, which includes a higher number of dual eligible older Medicare beneficiaries with disabilities who typically drive the highest medical cost expenditures as compared to other MSSP ACO cohorts and, in comparison, to national traditional Medicare beneficiary baselines. MSSP ACOs differ by constitution and characteristics, segmenting their Medicare beneficiaries, by severity: 1) ESRD 2) Disabled, 3) Aged Dual and 4) Aged Non Dual, averaging an annual average per capita spend of about $10,000 per beneficiary [7] Since M-ACO’s started in the MSSP on January 2013, the total attribution has tripled. Currently M-ACO has approximately 15,000 traditional Medicare and dual eligible attributables with a disproportionate share of the more costly dual and disabled beneficiaries that make up over 40% of all M-ACO attributed beneficiaries. Projected population growth rates in the M-ACO service area for those 50+ years of age is greater than double that of other age groups, and from the period 2000 until 2010, the 45–64 age cohort, or the “Baby Boomers”, experienced an increase of almost 50 percent; the largest growth of any age group analyzed by the U.S. Census for the M-ACO service region [7]. By 2040, those aged 65+ will reach over 1.5 million in the Atlanta region, more than the entire Georgia 65 + population as of this writing. The Atlanta region is cited by the Center for Disease Control as an area experiencing significant health disparities that are significantly influenced by social determinants of health [12].

M-ACO partners care for a disproportionate share of high-need, complex populations with evolutionary psychosocial burden and endure extraordinary challenges in managing utilization, with comparatively limited resources, calling for a different approach to improving cost and health outcomes. Organizational re-design that is implemented among M-ACO partners, including those that “treat and teach” in the community, is aimed to provide for a distinct implementation methodology to maximize broad and deep competence and accountability in managing such populations under new standards and metrics, with performance results that can be translated to other non-Medicare populations [9] According to the Health Resources and Services Administration (HRSA), 85% of the M-ACO service area is considered a medically underserved area (MUA) with several health professional shortage areas (HPSAs) including primary care, dental care, and behavioral health services [13].

### 2.4. Hypothesis

Our central hypothesis is that despite a disproportionate share of high-need patient population and the complexities associated with service fragmentation, the M-ACO health program redesign embedded in collaborative value-based performance principles, will achieve CMS targeted care, health, and cost objectives. 

### 2.5. Approach

M-ACO is guided by a strategic roadmap that reaffirms its shared goals and collaborative framework; based on the following 5 integrated success measures: 1) Promote financial viability that drives economic value to each ACO Participant organization to achieve positive net assets and operating margin to fund future advanced program enhancement and advanced technologies; 2) Enhance process improvement using evidence-based clinical and business practices; 3) Measurably improve service quality as defined by patients and their treatment plans (target improvement not less than 2% each year); 4) Maintain or improve physician and care team productivity that increase patients’ overall access to care as seamlessly as possible under a Patient Centered Medical Home (PCMH) delivery model; and 5) Support or improve clinical quality as defined by the M-ACO, CMS and other payers (Figure 1).

### 2.6. Implementing M-ACO Strategic Plan

Three distinct, and linked strategies are used to implement the shared goals: 1) Integrated and centralized care coordination; 2) web accessible communication platform; 3) centralized health information technology data warehouse with interoperability. The data warehouse aggregates data from multiple sources (electronic health records, health information exchange and state databases) that is funneled into the centralized risk stratification model which appoints patients to the right resources for care management. The work of the M-ACO includes ongoing deployment and comprehensive assessment of independent and collective clinical integration, health information technology capabilities, care coordination sophistication, service defragmentation, patient satisfaction and safety, governance and compliance strength, and the existing affiliations with community partners. Particular emphasis is placed on the unique population that the M-ACO and its community-based participant organization locations serve, the financing of their care, and the various structures and governance upon which they depend. The integration model developed by the M-ACO incorporate the education programs of Morehouse School of Medicine (MSM), its academic health center partner, which includes MSM’s primary care training capability, and its unique relationships among the M-ACO collaborators. In all respects, the application of benchmarks and standards are applied across all aspects of the organizational partners’ clinical and administrative operations.

### 2.7. Centralized (Integrated) Care Coordination

M-ACO serves a unique population in which more than half of the attributed Medicare beneficiaries have a disability or are eligible for Medicaid. Centralized care coordination (CCC) works to maximize clinical interventions by assisting with adherence to medications, treatment plans, and removing barriers to healthcare access [14,15,16,17].

Care Coordination was initially fragmented, and lacked a cohesive process, with each organization working independently. To serve a population that is potentially at high risk for complications, care coordination needed to be standardized with defined roles that required commitment from all parties involved. The M-ACO Centralized Care Coordination is based on the Patient Centered Medical Home (PCMH) (Figure 2). Standardization of care coordination begins at the point of care with the provider and care teams. Engaging the provider and health centers is just as important as engaging the patient. MCACO-ES discovered the biggest barrier to execution was related to the dissemination of information in a timely manner. Through the communication platform, specific campaigns and programs are provided through 1 to 3-minute short video clips with resources and toolkits available to all clinicians and care team.

The patient Centered Medical Home (PCMH) [14,15,16,17] is at the core of the patient experience, and underpins the Team based Centralized Care Coordination (CCC). Similarly, the team based care transitions and the specialist referral pathway, integrate with the PCMH framework. This integrated delivery model is enabled by the communications platform and the centralized health information technology data warehouse, which are described below (see Section 2.8 and Section 2.9).

### 2.8. Web Accessible Communications Platform

M-ACO’s evolving model of a web accessible Communication Platform includes focused education campaigns on Annual Wellness Visits, Chronic Care Management Analytics, Medication Therapy Management, and Centralized Care Coordination. M-ACO also utilizes a centralized care coordination management platform in which care coordination activities are documented and tracked. Tools rendered from an enterprise data warehouse and advanced analytics platform, high utilizers, modifiable population cohorts, cost, utilization and extensive referral patterns are identified for care coordination intervention.

The standardized process is directly linked within the care coordination management system so that all care teams have direct access to care coordination activities in a central platform. The centralized care coordination process is focused on connecting risk stratified patients with primary care providers to prevent complications for dual eligible and ambulatory sensitive conditions resulting in hospitalizations (See Table 2 and Table 3). The ultimate goal is to ensure patient-centered outcomes of centralized care coordination, including the use of mobile health technology, such as Health 360x in order to maximize clinical interventions of M-CACO providers by assisting with adherence to medications, treatment plans, and removing barriers to healthcare access [18,19,20].

This Data management pillar of the communication platform provides an evolving solution in working with multiple EMR systems within the ACO. The data received from various sources will go through a matching algorithm utilizing the EMPI methodology. The MCACO-ES data will flow in one direction from all known EMRs (i.e., NextGen, EPIC, E-Clinical Works, CMS and/or IDX), and be stored within a centralized Operational Data-Store (ODS). Once data is validated and aggregated, the summarized data will be moved into an additional database, which will allow for decision support and analytic reporting, when fully operationalized. 

### 2.9. Enterprise Data Warehouse with Interoperability

M-ACO informatics capability includes an integrated data warehouse, which supports Point of Care Data Integration, with HIPAA compliant data transfer from M-ACO participants and suppliers, through secure access to electronic medical records (EMR) and personal health information (PHI) data sources. The HIPAA compliant enterprise data warehouse and advanced analytics platform is configured to support the centralized care coordination and communications platform. As shown in Figure 3, the M-ACO data will flow in one direction from all known electronic medical records (EMRs) of participating organizations, including NextGen, McKesson Practice Partners, E-Clinical Works, EPIC, CMS, and/or IDX, and stored within a centralized Operational Data-Store (ODS). After data is validated and aggregated, the summarized data is moved into an additional database instance, which will allow for decision support reporting to the M-ACO senior management. After the data goes through a series of quantitative and qualitative quality control checks, and the reports are approved, the data is sub-populated into aggregates for PCMH, Quality Coalition, and Quality and Cost Metrics (including Meaningful Use and Medicare Shared Savings). 

### 2.10. Data Analysis

All data are reported as rates (percent or per 10,000 beneficiaries). Unit of analysis/summary was practice organization for Table 1, patients for Table 2, Table 3 and Table 4 and service for Table 5. Summary comparisons are shown between M-ACO and All Medicare Shared Savings Program (MSSP). 

## 3. Results

### 3.1. Medicare Beneficiary’s Health Conditions in M-ACO Compared with All Medicare Shared Savings Program (MSSP) ACOs

Medicare beneficiaries in MCACO-ES have higher rates of multiple complex conditions. As shown in Table 2, Medicare beneficiaries in M-ACO MSSP have 3–5 times higher rates of End Stage Renal Disease (ESRD) and Disabled patients (including Dual Eligible and Non-Dual Eligible). These patient populations as well as the Aged Dual Eligible patients have health challenges that are complicated by social determinants, such as low socioeconomic status and a zip code or place of residence that has limited access to health resources. By policy design, Aged Non-Dual Eligible Medicare patients do not qualify for Medicaid, and therefore have less complexities based on social determinants. This group comprises less than half of the Aged M-ACO population compared with 80% for the MSSP ACOs. Per the data set, All MSSP ACOs include averaged data from 400 ACOs participating under the MSSP. 

Similarly, M-ACO has higher rates per 10,000 beneficiaries of hierarchical chronic condition (CMS-HCC) that are associated with increased hospitalization and cost of care, such as diabetes with complications, chronic obstructive pulmonary disease, congestive heart failure, and major depression. (Table 3). M-ACO selected the most common chronic conditions (or HCCs) based on the volume of beneficiaries in comparison to the average MSSP ACO cohort. 

### 3.2. MSSP Shared Savings Outcomes for M-ACO

Despite high rates of multiple chronic conditions and health disparities, MCACO-ES generated shared savings for several performance years. For Performance Year 2015, MCACO-ES received $1,675,995 (based on a 50:50 sharing model with CMS). Shared savings is based on ACOs either meeting or remaining below the CMS-designated expenditures (or benchmark) per beneficiary. Another important factor includes triaging patients from the emergency room into primary care. Organizations that usually generate shared savings remain 5% below the benchmark cost reduction target and have positive risk adjustment factors. Risk adjustment factors are derived from patient demographics, such as age, gender and Medicare cohort and Medicare risk scores based on ICD-10 coding. M-ACO distribution of Shared savings:Proportion invested in infrastructure: 24%,Proportion invested in redesigned care processes/resources: 15%,Proportion of distribution to ACO participants: 61%.

### 3.3. M-ACO Compared with All MSSP ACO Quality Measures as Publicly Reported by CMS

M-ACO attained lower rates for several CMS Quality Measures, including four patient satisfaction measures, such as timely appointments and access to specialists and three quality measures including: influenza vaccination, and hypertension control (See Table 4). These quality measures along with twenty-three other measures are weighted and calculated by CMS to provide the M-ACO’s overall quality score, which was 92.06% in 2017.

### 3.4. M-ACO Access Primary Care Services and Specialist Physicians Compared with All MSSP ACO as Publicly Reported by CMS

The number of primary care visits in M-ACO have increased by 35.4% since 2016 compared to the CMS national benchmark which increased by 14% in the same time period (See Table 5). Increases in the number of primary care services are congruent with increases in overall cost of care, due to patients being diagnosed with chronic conditions in the primary care setting. The data also reflects increased emergency room and inpatient utilization as a result. Patients not getting into primary care does not impact whether shared savings are earned and can have a negative impact on cost containment, especially among high risk patients in M-ACO. 

## 4. Limitations

The operational efficiencies of the enterprise data warehouse remain a work in progress, primarily due cost of implementation and to non uniform adoption across organizations. 

Despite these challenges, the Communications Platform’s focused education campaigns on Annual Wellness Visits, Chronic Care Management Analytics, Medication Therapy Management, and Centralized Care Coordination are informing the M-ACO improvement process across practice organizations.

### 4.1. HCC Coding Opportunity

M-ACO HCC and Demographic risk scores are consistently lower, compared with CMS national means (see Table 6).

These data are informing 2019 HCC Coding Campaign (95% of Providers On-Boarded) and 37% HCC course completion, as well as 40% Annual Wellness Visit Completion, which increases billable care coordination management.

The data also informed care coordination targets for 2019: of 15,014 total attribution, 4725 are eligible for billable care coordination management (CCM); with a target set at 38% of the total eligible, and average expenditure of $8,715.23 per non ESRD beneficiary, the total target expenditure for shared savings is $83,903,729.00. 

### 4.2. Performance and Impact based on 1) Populations of Interest and 2) Processes and Outcomes of Care

While M-ACO has made significant progress in improving the quality of care, challenges remain with regard to access to specialists, as well as several CMS quality measures for preventive care, hospitalization, and diabetes control. M-ACO’s Centralized Care Coordination with use of patient engagement technology and telemedicine [18,19,20,21], are opportunities to address these challenges. Evaluation for ongoing assessment of M-ACO’s Care Coordination Agency Model tests the effectiveness of the Care Coordination Agency Model (control group) compared to those cohorts that do not receive care coordination (experiment group). Successful improvement strategies shown to improve patient health outcomes will measure the operational cost to provide care coordination services compared to the savings generated from improving health outcomes for patients with multiple chronic conditions. The model with optimal achievement will be deemed as the standardized care coordination protocol, which will be used to train care coordinators across M-ACO practices. Morehouse School of Medicine will assist with the development of dashboards to measure, track and analyze data collected from both models. Potential clinical impact of this approach among high risk ACO patients with diabetes from a single health center was recently demonstrated, using a sample of 41 patients in a NIH funded pilot study: 19 were diagnosed with Type 2 diabetes. 58% (11 patients) reported decreasing A1C since enrollment into centralized care coordination with a notable reduction in A1C from 13.5% to 6.8% within a span of only 3 months [21]. This original research was sponsored by the National Institutes on Minority Health and Health Disparities (NIMHD) at NIH, through funding for the Transdisciplinary Collaborative Center (TCC) for Health Disparities Research at Morehouse School of Medicine under the leadership of Dr. David Satcher, Founder and Senior Advisor of the Satcher Health Leadership Institute and 16th Surgeon General of the United States. The TCC highlights collaborative and health policy innovations that address upstream and modifiable risks to attaining health equity [22,23,24,25]. The M-ACO will continue to explore and prioritize such interventions for testing and scaling across the ACO and similar practices serving high risk patient populations, including dual eligible Medicare beneficiaries.

In describing the framework for evaluating ACOs, Fisher et al [26] note the importance of tracking the impact of the accountable care models on subgroups of the population at greater risks, such as socioeconomically disadvantaged populations or people who are cared for by safety-net providers, anticipating greater challenges of meeting target savings or quality outcomes among such disadvantaged and high risk patients. Processes and outcomes of care such as patients’ experience of care, including degree of activation, engagement in shared decision making, and decision quality are especially important in the evaluation framework [26]. Inclusion of such populations and patient experience outcomes are not generally described across ACOs [26,27].

### 4.3. Future Directions to address Limitations in Data Integration, Performance and Impact

Data integration across disparate EMRs remain a challenge across healthcare. Recent technology advancement with FHIR (Fast Healthcare Interoperability Resource) enables apps to connect to electronic health record systems [28,29]. Even as it continues to implement its enterprise data warehouse integration model, the M-ACO is exploring app enabled FHIR EMR accessibility using a patient engagement platform that is currently used by ACO member patients to monitor and track their health [18,20,21]. Such patient engagement application will enable data collection on patient experience and shared decision making, a critical dimension for evaluation and care outcomes.

## 5. Discussion

Based on the MSSP model, the project objectives focused on 1) Promoting financial viability 2) Enhancing process improvement 3) Measurably improving service quality 4) maintaining or improving physician and team productivity 5) Improving quality as defined by CMS. Our findings show that despite high rates of multiple chronic conditions and health disparities, M-ACO, is delivering quality care and has achieved incentive payments in the Medicare Shared Savings Program. Given constant changes in the financing of health care services, M-ACO remains ready and committed to the mission of improving population health through evidence-based clinical and administrative models that prioritize the patient experience, while addressing social determinants and behavior risk modification. Already, M-ACO’s strategic approach has improved the health status of its attributed population and it has been financially rewarded for such outcomes through shared savings incentive payments. 

The value and contribution of this study is the unique opportunity to describe MSSP outcomes in a large safety net and predominant primary care practice of a diverse Medicare beneficiary patient population. 

A comparison of MSSP ACO cohorts from 2012 and 2013 showed savings in the earlier cohort of primary care based MSSP, and less so for hospital based MSSP [27] However, these cohorts were less diverse with 82% white beneficiaries, and only 8.2% and 5.1% Black and Hispanic beneficiaries respectively; and only 8.7% recipients below the federal poverty level [27].

Future and pursuant work will focus on empirical validation of the summary comparisons presented in this manuscript by expanding data to allow inferential analyses that will employ among other approaches statistical methods like random effects and propensity scores to minimize bias and control for confounders. 

## 6. Conclusions

M-ACO will continue to seek ways to innovate care delivery, as we expand our practice geographic area, to ultimately deliver care to 100,000 beneficiaries with expanding coverage to rural and urban communities across Georgia. The Centralized Care Coordination model with attention to preventive care, care transitions and referral pathways are key targets for innovation. Integration of mobile health technology for patient engagement and retention, as well as telemedicine technology, will be tested to determine how these technologies improve quality care in the high-risk patients with complexities and social determinants, that we serve at the Morehouse Choice Accountable Care Organization and Education System. 

## Figures and Tables

**Figure 1 ijerph-16-03084-f001:**
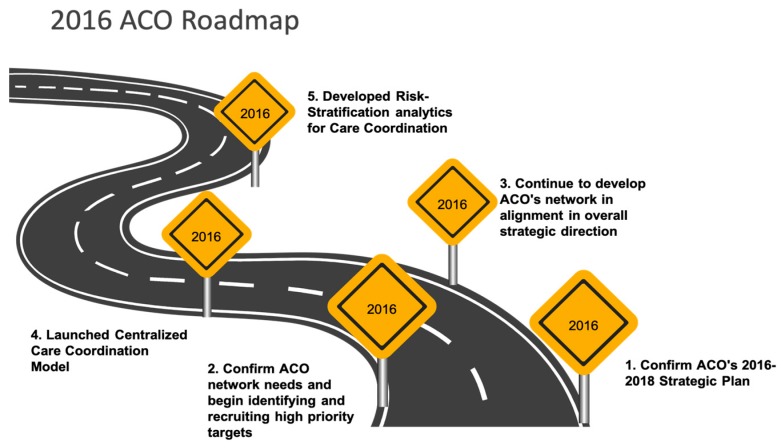
2016 M-ACO Roadmap Reaffirms Strategic Approach and Confirms Strategic Plan (2016–2018).

**Figure 2 ijerph-16-03084-f002:**
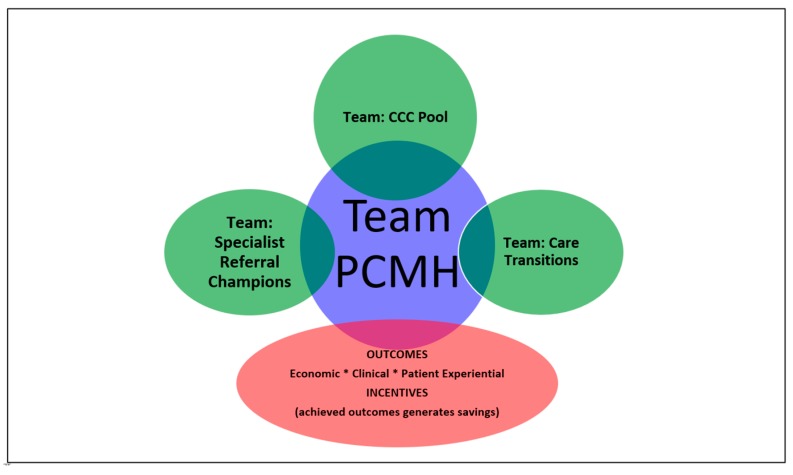
Integrated Care Delivery Model and Centralized Care Coordination.

**Figure 3 ijerph-16-03084-f003:**
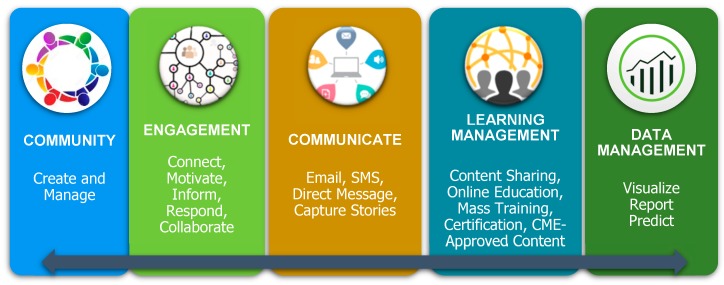
Communication Platform Pillars: The 5 Components of M-ACO Communication Model are Community; Engagement; Communicate; Learning Management; Data Management.

**Table 1 ijerph-16-03084-t001:** M-ACO Participant Organizations and Health Center Locations Across Urban and Rural Georgia.

Organization	Practice Type and Locations
Morehouse School of Medicine and Morehouse Healthcare	Primary care and select multispecialty: Two Urban locations
Southside Medical Center	FQHC: Eleven Urban and Rural locations including school-based health centers, Adult and Pediatric Primary care, specialty care, dental optometry, behavioral health and mobile medical and dental unit.
Family Health Centers of Georgia	FQHC: Seven Urban locations including school-based health centers, Adult and Pediatric Primary care, specialty care, dental optometry, behavioral health and mobile medical and dental unit.
Four Corners Primary Care	FQHC: Three Urban locations
CareConnect Health	FQHC: Forty-one Rural locations, including school-based clinics, Dental, OB/GYB, Urgent Care Centers.
Medical Associates Plus	FQHC: Eight Suburban and Rural locations including Adult and Pediatric Primary care, specialty care, dental optometry, behavioral health and pulmonary health
Community Health Care Systems	FQHC: Thirteen Rural locations including Adult and Pediatric Primary care, podiatry, behavioral health and mobile medical unit
East Georgia Healthcare Center	FQHC: Ten Rural locations including Adult and Pediatric Primary care, specialty care, dental and behavioral health
MedLink Georgia	FQHC: Eighteen Rural locations including Adult and Pediatric Primary care, specialty care, dental optometry, behavioral health
Albany Area Primary Health Care	FQHC: Twenty-six Rural locations including school-based health centers, Adult and Pediatric Primary care, specialty care, dental optometry, behavioral health and mobile medical and dental unit.
North Georgia Healthcare Center	Independent Rural practice including Adult and Pediatric Primary care, specialty care, dental optometry, behavioral health and physical therapy
Atlanta Family Physicians	Independent Urban practice
The Clinic For All	Independent Urban Practice

**Table 2 ijerph-16-03084-t002:** Medicare Beneficiary Health Conditions (%) in MCACO-ES Compared to All MSSP ACOs.

Medicare Beneficiary	MCACO-ES *	All MSSP ACOs
ESRD	2.96%	0.69%
ESRD Dual Eligible	1.34%	0.24%
ESRD Non-Dual Eligible	1.62%	0.45%
Disabled	33.36%	11.72%
Disabled Dual Eligible	17.61%	5.56%
Disabled Non-Dual Eligible	15.75%	6.14%
Aged	63.68%	87.41%
Aged Dual Eligible	15.99%	6.19%
Aged Non-Dual Eligible	47.69%	80.86%

* *p* < 0.001.

**Table 3 ijerph-16-03084-t003:** Frequencies and Rates per 10,000 Beneficiaries by Disease Group (CMS-HCC) for Assigned Beneficiaries.

CMS-HCC Condition	MCACO-ES	All MSSP ACOs
Diabetes w/Chronic complications	2134	1670
Chronic Obstructive Pulmonary Disease	1481	1318
Congestive Heart Failure	1202	1196
Morbid Obesity	1180	544
Diabetes without Complication	1162	1172
Vascular Disease	1120	1402
Major Depressive, Bipolar & Paranoid Disorder	930	737
Specified Heart Arrhythmias	898	1525
Rheumatoid Arthritis and Inflammatory Connective Tissue Disease	644	716
Seizure Disorders and Convulsions	500	295

**Table 4 ijerph-16-03084-t004:** Select ACO Quality Measures as publicly reported by CMS.

Select ACO Quality Measures	M-ACO (%)	All MSSP ACOs (%)
Getting Timely Care, Appointments, and Information	72.20	80.60
How Well Your Providers Communicate	90.71	93.13
Access to Specialists	78.43	83.32
Shared Decision Making	70.36	75.85
Influenza immunization	62.37	72.52
Diabetes A1c poor control	23.94	16.74
Hypertension control (High BP control)	60.18	71.47

**Table 5 ijerph-16-03084-t005:** Primary Care Services as publicly reported by CMS. FQHC/RHC=Federally Qualified/Rural Health Clinic; FFS=Fee for Service.

Primary Care Services	M-ACO	All MSSP ACOs	National Assignable FFS
With a Primary Care Physician	8760	9711	10,120
With a Specialist Physician	1004	3811	3713
With a Nurse Practitioner/Physician Assistant/Clinical Nurse	3383	4348	4491
With a FQHC/RHC	762	1110	1427

**Table 6 ijerph-16-03084-t006:** HCC and Demographic Risk Scores by Medicare Category in M-ACO HCC compared with CMS (National Means).

Medicare Category	M-ACO HCC Risk Score	CMS-HCC Risk Score (National Mean)	M-ACO Demographic Risk Score	CMS Demographic Risk Score (National Mean)
ESRD	0.893	1.115	1.005	1.021
Disabled	0.865	1.282	0.987	1.057
Aged/Dual	0.750	1.805	0.946	1.562
Aged/Non-Dual	0.904	1.055	1.002	0.911

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
