# Peer review of "Morehouse Choice Accountable Care Organization and Education System (MCACO-ES): Integrated Model Delivering Equitable Quality Care"

_ijerph, 2019, doi:10.3390/ijerph16173084_

Round 1

Reviewer 1 Report

The authors attempted to present their research approach, early outcomes, lessons learned, and future directions for this physician-led and community-focused ACO. I think the authors have been quite successful in achieving their objectives. However, some important section is still missing.

In particular, in the Discussion section authors must compare their article with similar ones, or with others with similar objective, or topic, etc. 

They must also include a paragraph or two of limitations.

Author Response

Comments and Suggestions for Authors

The authors attempted to present their research approach, early outcomes, lessons learned, and future directions for this physician-led and community-focused ACO. I think the authors have been quite successful in achieving their objectives. However, some important section is still missing.

We thank the Reviewer for this thoughtful review and feedback.

Comment: In particular, in the Discussion section authors must compare their article with similar ones, or with others with similar objective, or topic, etc. 

Response: Please see updated discussion: lines 348-353; 359-365.

Comment: They must also include a paragraph or two of limitations.

Response: Please see section 4.0: lines 311-346

Reviewer 2 Report

This is a very informative paper on ACO performance.  The case study approach reveals the descriptive nature of the study design and analysis. Information presented is helpful to the general public, but it fails short in several areas.

  Hypothesis:  The hypothesis is not empirically validated although a testable hypothesis was presented in the paper.

 Data Analytics:  It is unclear how patient care outcomes were analyzed or examined although the CMS database could reveal the differentials in ACO performance by the type of ACOs.

Simulation:  It is unclear about how simulation is conducted.  

Unit of Analysis: It is unclear about the unit of analysis of the project. It is important to point out the contextual differences of different ACO operations.  The propensity score matching and analysis could be an excellent solution of biased selection when multiple ACOS are compared.

In conclusion, this is an important project funded by NIMHD.  Information presented does not show much of empirical results on the impact of coordinated care or integrated delivery system.  Several ACO related evaluation articles appeared in Health Services Research and Managerial Epidemiology and Health Care Management Science could be good sources for citations.  

Author Response

Comments and Suggestions for Authors

We thank the reviewer for the thoughtful suggestions and feedback.

This is a very informative paper on ACO performance.  The case study approach reveals the descriptive nature of the study design and analysis. Information presented is helpful to the general public, but it fails short in several areas.

  Hypothesis:  The hypothesis is not empirically validated although a testable hypothesis was presented in the paper.

Response: We have updated the outcomes and addressed limitations

 Data Analytics:  It is unclear how patient care outcomes were analyzed or examined although the CMS database could reveal the differentials in ACO performance by the type of ACOs.

Response: Please see section 4.0: lines 311-346. Data has been collected using the Communications Platform described, as well as data from CMS on outcomes.

Simulation:  It is unclear about how simulation is conducted.

Response:  The Data enterprise remains a work in progress as described above.

Unit of Analysis: It is unclear about the unit of analysis of the project. It is important to point out the contextual differences of different ACO operations.  The propensity score matching and analysis could be an excellent solution of biased selection when multiple ACOS are compared.

Response: We have included unit of analysis under 2.10 data Analysis: lines 256-258. It was not our intent to compare ACOs. We have described Morehouse Choice ACO in the context of the Medicare Shared Savings Program (MSSP). In this context, the patient demographics and risk scores are relevant to outcomes, as described in the expanded and updated discussion: lines 348-353; 359-365.
